# Effect of Degradation of Polylactic Acid (PLA) on Dynamic Mechanical Response of 3D Printed Lattice Structures

**DOI:** 10.3390/ma17153674

**Published:** 2024-07-25

**Authors:** Reza Hedayati, Melikasadat Alavi, Mojtaba Sadighi

**Affiliations:** 1Faculty of Aerospace Engineering, Delft University of Technology (TU Delft), Kluyverweg 1, 2629 HS Delft, The Netherlands; 2Department of Mechanical Engineering, Amirkabir University of Technology, Tehran P.O. Box 15875-4413, Iran; melika.alavi@aut.ac.ir (M.A.); mojtaba@aut.ac.ir (M.S.)

**Keywords:** 3D printing, PLA degradation, material extrusion, low-velocity impact, sandwich panels

## Abstract

Material-extrusion-based 3D printing with polylactic acid (PLA) has transformed the production of lightweight lattice structures with a high strength-to-weight ratio for various industries. While PLA offers advantages such as eco-friendliness, affordability, and printability, its mechanical properties degrade due to environmental factors. This study investigated the impact resistance of PLA lattice structures subjected to material degradation under room temperature, humidity, and natural light exposure. Four lattice core types (auxetic, negative-to-positive (NTP) gradient in terms of Poisson’s ratio, positive-to-negative (PTN) gradient in terms of Poisson’s ratio, and honeycomb) were analyzed for variations in mechanical properties due to declines in yield stress and failure strain. Mechanical testing and numerical simulations at various yield stress and failure strain levels evaluated the degradation effect, using undegraded material as a reference. The results showed that structures with a negative Poisson’s ratio exhibited superior resistance to local crushing despite material weakening. Reducing the material’s brittleness (failure strain) had a greater impact on impact response compared to reducing its yield stress. This study also revealed the potential of gradient cores, which exhibited a balance between strength (maintaining similar peak force to auxetic cores around 800 N) and energy absorption (up to 40% higher than auxetic cores) under moderate degradation (yield strength and failure strain at 60% and 80% of reference values). These findings suggest that gradient structures with varying Poisson’s ratios employing auxetic designs are valuable choices for AM parts requiring both strength and resilience in variable environmental conditions.

## 1. Introduction

Additive manufacturing (AM), particularly fused deposition modeling (FDM), has revolutionized the design and production of intricate lattice structures. Before the advent of AM, these complex geometries presented significant challenges in design and manufacturing [1]. Lattice structures offer a unique combination of advantageous properties, including a high strength-to-weight ratio, light weight, and the ability to tailor microstructural characteristics for specific applications in mechanical engineering [2,3,4,5,6].

One particularly exciting class of lattice structures for use in sandwich panels is auxetics [7]. These exotic and fascinating materials exhibit a counterintuitive property: negative Poisson’s ratio. In contrast to conventional materials that contract laterally when stretched, auxetics expand laterally under tensile loading [8]. This unique property makes auxetic lattices highly valuable for various applications, particularly those requiring high impact resistance and energy absorption [9], in biomedical engineering, vehicle components, and sports equipment [10,11,12].

Among various 3D printing techniques, FDM has received significant research attention [13,14,15], particularly because of its potential to utilize sustainable materials. Polylactic acid (PLA), a bio-derived thermoplastic known for its biodegradability, biocompatibility, mechanical strength, and ease of processing, serves as a compelling alternative to traditional materials across various applications [16,17,18].

The combination of FDM and PLA (FDM/PLA) printing emerges as a promising approach for producing porous and lattice structures due to FDM’s cost-effectiveness, accessibility, and design flexibility, and PLA’s eco-friendly nature and remarkable mechanical properties [19].

However, understanding PLA’s long-term mechanical performance under environmental effects is crucial for determining its suitability in various applications. Like many polymeric materials, PLA is susceptible to degradation and the loss of mechanical properties over time. Several factors contribute to PLA degradation, including thermal decomposition, hydrolysis, photo-oxidation, and natural weathering [20,21].

These processes decrease PLA’s molecular weight, shorten polymer chains, and ultimately weaken the overall structure, particularly in terms of strength, stiffness, and impact resistance [22,23,24,25]. Moisture absorption is a critical concern for 3D printed PLA structures, as water molecules break down the ester bonds of the polymer chain, accelerating degradation [26] and increasing brittleness [27,28]. Porous PLA structures, with their increased exposed surface area, are particularly prone to moisture absorption compared to solid PLA. Additionally, temperature fluctuations exacerbate degradation, with higher temperatures accelerating the process even at slower rates at room temperature [29,30].

Current research on PLA degradation often focuses on quasi-static properties under controlled conditions. This approach neglects the complex interplay between real-world environmental factors and the dynamic response of PLA porous structures, particularly lattice structures employed in sandwich panels. These lightweight structures, primarily designed for high energy absorption capability and impact resistance, are highly susceptible to atmospheric degradation.

This study addresses the knowledge gap by investigating the time-dependent effects of environmental factors on the quasi-static and particularly dynamic mechanical properties of PLA-based lattice structures in sandwich panels. We assess degradation by testing the bulk material mechanical properties of PLA at different aging stages (as-printed, 45 days, 90 days). Subsequently, we analyze the resulting impact behavior of sandwich panels with varying core topologies (conventional honeycomb, auxetic, and cores with graded Poisson’s ratio) subjected to simulated reductions in key material properties (yield strength, failure strain) at different levels (20%, 40%, 60%). Understanding this degradation process will ultimately inform the design of more durable and reliable PLA-based sandwich panels for real-world impact scenarios.

## 2. Materials and Methods

### 2.1. Experimental Tests

#### 2.1.1. Manufacturing

To produce sandwich panels, a widely used additive manufacturing technique FDM was employed. Specimens with four distinct core geometries were 3D printed: conventional honeycomb, auxetic, negative-to-positive (NTP) gradient, and positive-to-negative (PTN) gradient (Figure 1). Three specimens were manufactured for each type of sandwich panel. For the NTP gradient and PTN gradient core designs, the Poisson’s ratio of the core layers transitioned from negative to positive and positive to negative, respectively, along the thickness. This variation was achieved by manipulating the internal angle of the unit cells. All sandwich panels were fabricated from PLA using a Creality Ender 3-Pro 3D printer and possessed identical dimensions of $9\times 9\times 4.3{\;\mathrm{cm}}^{3}$.

#### 2.1.2. Tensile and Compression Tests

To assess how environmental exposure affects the mechanical properties of PLA, tensile and compression tests (Figure 2) were performed on specimens aged for different durations: as-printed, 45 days after production, and 90 days after production. These tests aimed to quantify the influence of aging on two key material properties: failure strain and yield strength. Both properties are crucial for a material’s ability to absorb energy and resist deformation. Lower failure strain indicates a more brittle material with a reduced capacity to absorb impact and redistribute stress, leading to a higher risk of sudden failure. Additionally, brittle materials exhibit lower tolerance to external forces and environmental degradation. Lower yield strength can also negatively impact structural response, even under low-velocity impacts. A lower yield point can lead to increased deformation, localized stress concentrations, and progressive damage accumulation over time.

The testing employed standard dogbone specimens for tensile tests (dimensions provided in the Online Appendix A) and cylindrical specimens (with a diameter of 12.7 mm and a height of 25.4 mm) for compression tests. A constant displacement rate of 1mm/min was used for the tests, which involved measuring the elastic modulus, yield stress, and failure strain of PLA. To ensure consistent results and to replicate realistic exposure scenarios, the specimens were kept under controlled environmental conditions throughout aging and testing. The temperature ranged from 5 to 15 °C, and relative humidity was maintained around 70%. Direct sunlight exposure was avoided, resulting in a quite low UV index (<2). Notably, no environmental factor was extremely elevated or decreased, and the environmental factors were maintained within real-world ranges.

#### 2.1.3. Drop-Weight Impact Tests

Drop-weight impact testing was employed to validate the comprehensive numerical results obtained in this study. A hemispherical-ended cylindrical impactor with a 16 mm diameter and 50 mm length was utilized. The impactor was positioned 60 cm above the top surface of each panel, resulting in an initial impact velocity of 3.43 m/s. A fully clamped boundary condition was used, and the weight of the cylindrical impactor and extra weights on it was 2.7 kg. The sandwich panels were tested after manufacturing, so the material was considered undegraded.

### 2.2. Numerical Modeling

To investigate the effects of environmental exposure on 3D printed PLA sandwich panels under low-velocity impacts, numerical simulations were conducted. These simulations explored a comprehensive range of material properties for all four core geometry types (honeycomb, auxetic, NTP gradient, and PTN gradient). Sixteen distinct combinations of yield strength and failure strain were investigated. For each combination, the yield strength and failure strain of the PLA material were systematically adjusted within a defined range (100%, 80%, 60%, and 40%) relative to the reference values obtained from quasi-static tests on as-fabricated specimens. This analysis aims to explore the degradation mechanisms affecting these key mechanical properties (failure strain and yield strength), allowing us to evaluate the impact of environmental exposure on the integrity and performance of structures during low-velocity impacts.

The finite element (FE) analysis comprised two stages: geometrical design and subsequent simulations. The ANSYS APDL code was used for geometric design, and the LS-DYNA 971 code was employed for the numerical simulations.

To efficiently capture the in-plane behavior of the lattice structure, two-dimensional (2D) shell elements were used for the sandwich panel, while 3D solid elements were employed for the impactor, the additional weight, and the supporting plate. The core material was assigned a bilinear isotropic material model (plastic kinematic model in LS-DYNA) to capture its ability to undergo both elastic and plastic deformations. A simpler linear isotropic material model was chosen for the impactor, while both the added weight and supporting plate were modeled as rigid materials due to their negligible deformations during the impact event.

A uniform element size of 3 mm was employed to discretize both the sandwich panel and the added weight, while a finer element size of 1.5 mm was used for the impactor. The support plate was assigned material properties identical to those of the added weight for simplicity. The material properties for all components, which can be seen in Figure 3, are detailed in Table 1.

The initial velocity was set to 3.43 m/s, mimicking the experiments. The implemented contact types included node-to-surface contact for the interaction between the core and the impactor and automatic general contact for the surfaces of the cell walls in the core.

To improve the accuracy of the results, hourglass control parameters were employed. These parameters regulate hourglass energy. IHQ and QH were set to 2 and 0.14, respectively. Additionally, Q1 and Q2 were set to 2 and 0.25 for bulk viscosity control. These parameters were carefully chosen based on established practices to optimize the simulation and enhance the results’ quality.

## 3. Results

### 3.1. Changes in the Bulk Material Properties

As expected, tensile tests revealed a decrease in failure strain for PLA over time (Figure 4a). Specimens tested 45 days after manufacturing exhibited a mean failure strain of 0.036, a 5.3% reduction compared to the reference value of 0.038 measured for specimens tested just after manufacturing. Both ultimate strength and yield strength followed a similar trend, decreasing by 6.9% and 6%, respectively, reaching 39.3 MPa and 39 MPa from their reference values of 42.2 MPa and 41.5 MPa for specimens tested 45 days after printing. This trend became more pronounced for specimens tested after 90 days. Their mean failure strain further declined to approximately 0.032, representing a substantial 16% decrease compared to the reference condition. The ultimate strength also exhibited a significant decline, reaching close to 36.1 MPa, showcasing a 14.5% decrease relative to the reference value.

Compared to compression, the aging process had a more significant impact on tensile properties, especially on failure strain, which showed a steeper reduction over time (Figure 4a). Notably, during the second 45 days of aging, there was an 11.1% reduction in failure strain, approximately twice the reduction observed during the initial 45-day period.

While the walls of the sandwich structures primarily experienced tensile stress under impact conditions (based on the observations in our numerical studies), the influence of aging on compressive properties was also investigated. As illustrated in Figure 4b, the yield strain remained relatively unaffected even after 90 days of aging, remaining at approximately 0.057. However, the elastic modulus and consequently the yield strength were more susceptible to degradation. The elastic modulus decreased from 1.6 GPa to nearly 1.5 GPa, reflecting a 6.25% decline.

### 3.2. Drop-Weight Behavior of Undegraded PLA Sandwich Panels

To validate the FE model for predicting the impact behavior of PLA sandwich panels, drop-weight impact tests (Figure 5) were conducted on the reference sandwich panels. The results, including displacement–time and kinetic energy–time curves (Figure 6) of the impactor penetrating the core, demonstrated good agreement between the FE model and experiments.

### 3.3. Parametric Study

The influence of environmental exposure on the stress distribution within the structures was investigated through an analysis of von Mises stress (Figure 7). It was observed that, in general, the structures exhibited comparable peak stress values, particularly during the initial stages of impact, regardless of the core type design. Despite some variations in the trends observed in the reference condition (e.g., auxetic showed a significant decline compared to the other cores), it is worth noting that the influence of bulk material degradation stage in changing the levels of maximum von Mises stress was noticeably higher than the effect of core geometry types.

A visualization of von Mises contours in auxetic structures (Figure 8) and honeycomb structures (Appendix A) is provided.

Cross-sectional views of the impacted region of the auxetic structure and the volume fraction of the eroded elements in all structures are demonstrated in Figure 9 and Figure 10, respectively (see also the cross-sectional view of the impacted region of the honeycomb structure in Figure A1 of Appendix B). These graphs provide valuable insights into the damage form and extent sustained by the walls of these cellular structures. As expected, as compared to honeycomb structures, the material was more concentrated in the vicinity of the impact point in the case of the auxetic structures (compare Figure 9 and Figure A1).

Figure 10 illustrates a clear correlation between the level of wall rupture and the degradation level. As the yield strength and failure strain decrease, the damage becomes more pronounced. For example, in the undegraded condition, the fraction of eroded elements was 0.007, 0.012, 0.01, and 0.0098 in auxetic, honeycomb, NTP gradient, and PTN gradient structures, respectively. However, after a reduction in yield strength and failure strain to 60% of their initial values, the noted fractions increased by 2.86, 2.1, 2.3, and 1.9 times, respectively, in the mentioned cores. Therefore, after weakening the failure strain and yield stress by 40%, the PTN gradient structure showed the least amount of erosion, while the honeycomb structure exhibited the highest. This highlights the superior damage tolerance of structures incorporating cells with negative Poisson’s ratio. Additionally, it showcases the benefits of using gradient structures in degradation-prone situations.

One of the primary applications of sandwich structures with cellular cores is energy absorption during impacts. Therefore, comparing the energy absorption capability values is a key aspect in evaluating their performance. Figure 11 presents the internal energy absorbed by auxetic, NTP gradient, PTN gradient, and honeycomb cores under varying yield stress and failure strain conditions. A key finding is that brittleness significantly reduced energy absorption across all core types. Structures with the highest employed yield strength but a low failure strain (40%, indicating relative brittleness) exhibited an energy absorption reduction even exceeding two-fold. This effect was more pronounced in honeycomb structures, where the reduction reached a three-fold reduction compared to auxetic structures (around a 1.5-fold reduction). Interestingly, there was a positive correlation between impactor penetration depth and energy absorption, except for brittle conditions. This implies that deeper penetration allowed for more energy dissipation throughout the structures. Notably, in structures with non-brittle material (100% failure strain), the level of yield stress had minimal impact on energy absorption. However, structures with lower yield strength required a longer time to reach their peak internal energy.

Interestingly, under reference conditions (i.e., 100% yield stress and failure strain), gradient structures exhibited enhanced energy absorption capabilities, exceeding 5 J (Figure 11). This surpassed the values absorbed by both auxetic and honeycomb structures. As expected, honeycomb cores displayed the lowest energy absorption capacity among all core types. See a more concise visualization of internal energy focusing solely on the highest and lowest failure strain levels in Appendix A.

Generally, the peak force on each force–displacement curve represents the core’s resistance to the impact. A key finding is the significant influence of brittleness on peak force, particularly in auxetic structures (Figure 12). Decreasing the failure strain at the highest yield strength resulted in a more substantial reduction in peak force compared to simply decreasing the yield stress while maintaining a high failure strain (100%). This suggests that brittleness plays a more critical role than yield strength in determining peak force value.

For the auxetic core, peak force decreased significantly when both failure strain and yield strength were reduced. For example, at the highest failure strain (100%), the peak force dropped from 1075 N to 900 N as yield strength decreased by 60% (Figure 12a). On the other hand, reducing failure strain from 100% to 40% at the highest yield strength (i.e., $\sigma_{ys}\;100\%$) resulted in a more substantial drop (1075 N to 500 N). A similar analysis of other core types confirms that brittleness plays a more significant role in determining peak force (Figure 12).

For the same bulk material properties, the structures with pure auxetic cores (Figure 12a) exhibited higher levels of peak force as compared to structures with pure honeycomb cores (Figure 12d). However, the length of the plateau region in the structures with honeycomb cores was higher than that of their auxetic counterparts, especially for constituent weaker material (lower yield strength and failure strain). This makes gradient structures a more logical choice when a structure needs to absorb energy while maintaining sufficient strength over time.

Consider structures with bulk material properties at 60% and 80% of the reference values for yield strength and failure strain, respectively. In this scenario, none of the impactors hit the plate. The peak forces for the auxetic, NTP gradient, PTN gradient, and honeycomb cores were 840 N, 682 N, 800 N, and 503 N, respectively. While the NTP gradient core allowed slightly higher overall displacement (35 mm vs. 31 mm for auxetic), the peak force resistance remained comparable (around 800 N vs. 840 N for auxetic).

On the other hand, the energy absorption curves in Figure 11 show that the final energy absorption for the gradient cores (3.49 J–3.6 J) was higher than that in the auxetic core (2.5 J) for this case. This example highlights the ability of gradient structures to create a balance between energy absorption capability and deformation resistance for the majority of structures with both intact and degraded bulk constituent materials. The force–displacement curves corresponding only to the highest and lowest failure strain levels are provided in Appendix A.

Velocity–time curves (Figure 13) provide valuable insight into the residual velocity of the impactor, obtained at the end of the curves, which serves as an indicator of energy dissipation and absorption by the core. In structures with the highest level of failure strain (i.e., where $\varepsilon_{f}$ is equal to the reference value), the return velocities ranged from −1 m/s to −0.5 m/s for all core types except the auxetic structure with low yield strength (40% and 80% of reference). In these two specific auxetic cases, the residual velocity magnitudes surpassed 1 m/s (Figure 13a).

Interestingly, across all structures, the residual velocity magnitude for structures with 80% yield strength equaled or exceeded that of structures with 100% yield strength. Nonetheless, decreasing yield strength to values lower than 80% usually led to lower residual velocities. When examining the influence of brittleness, increasing the brittleness usually led to lower levels of residual velocity. For structures with 100% yield strength, decreasing the failure strain from 100% to 60% resulted in a change in residual velocity from −0.93 m/s to −0.74 m/s, −1.03 m/s to −0.71 m/s, −0.94 m/s to −0.29 m/s, and −0.73 m/s to −1.81 m/s in the auxetic, NTP gradient, PTN gradient, and honeycomb structures, respectively. It must be noted that the abrupt decline in the velocity (for instance at t=16 ms in the auxetic structure and at t=15 ms in the honeycomb structure) is related to the impactor hitting the back support plate.

To compare the impact of the yield stress of the auxetic core on the displacement of the impactor, we first consider structures with identical failure strains of 100%. The core with the lowest yield strength exhibited a displacement of 27 mm, approximately 24% greater than that of the strongest material (Figure 14a). For conventional honeycomb cores with the same failure strain, the structure with the weakest yield strength had a displacement around 9 mm greater than the one with the highest yield strength. This difference represents a 36% increase, larger than the change seen in auxetic cores (Figure 14d).

Auxetic structures generally exhibited lower maximum displacements compared to their honeycomb counterparts (Figure 14a,d). The gradient structures are expected to exhibit behaviors between those of auxetic and honeycomb structures. Interestingly, in the undegraded structures, the maximum displacement of the gradient structures (25 mm and 26.1 mm) was higher than those in the auxetic and honeycomb structures (21.8 mm and 23.3 mm). Nonetheless, in the degraded structures, the maximum displacement of the gradient structures fell between the values of the auxetic and honeycomb structures in most cases.

As expected, lower yield stress resulted in increased maximum displacement. Reducing the yield stress from 100% to 40% (with the failure strain remaining at the original value) increased the maximum displacement by 23.4% (from 21.8 mm to 26.9 mm), 32% (from 25 mm to 33 mm), 21.8% (from 26.1 mm to 31.8 mm), and 35.2% (from 23.3 mm to 31.5 mm) in the auxetic, NTP gradient, PTN gradient, and honeycomb structures, respectively. Notably, for undegraded structures, the NTP gradient structure exhibited lower displacement compared to the PTN gradient structure at 40% and 60% yield stress, but the opposite occurred at higher yield strengths.

Decreasing the failure strain from 100% to 60% (with constant yield stress at its reference value) resulted in significant displacement increases by 42.2% (from 21.8 mm to 31 mm), 56% (from 25 mm to 39 mm), 71.3% (from 26.1 mm to 44.7 mm), and 95% (from 23.3 mm to 45.4 mm) in the auxetic, NTP gradient, PTN gradient, and honeycomb structures, respectively. This trend held true even for structures with an intermediate yield stress of 60%. At a failure strain of 40%, the impactor penetrated the entire thickness of all lattice structures, reaching the supporting plate (as shown in Figure 14).

Overall, failure strain had a more significant effect on displacement than yield stress. This is further evidenced by the auxetic structure, where lowering the failure strain from 100% to 40% with the highest yield stress doubled the displacement, while reducing the yield stress from 100% to 40% with the highest failure strain only decreased displacement by 24% (Figure 14a).

Interestingly, the auxetic and gradient structures exhibited less susceptibility to degradation compared to the conventional honeycomb structure. As shown in Figure 15, undegraded gradient structures exhibited greater displacement than both auxetic and honeycomb structures (the right side of Figure 15). Following degradation, the displacement values of the gradient structures fell between those of the auxetic and honeycomb structures (the left side of Figure 15). Therefore, incorporating both negative and positive cells inside the lattice structure made the structure more resilient to the degradation of bulk material.

## 4. Discussion

### 4.1. Prevalence of PLA 3D Printed Products

PLA is one of the most commonly used materials for production with the 3D printing method due to its affordability, ease of use, and environmental friendliness. In industries like medical implants, telecommunications, electronics, and aerospace, there is a growing trend towards the use of additive manufacturing methods like FDM for creating high-quality parts. FDM offers advantages such as simplicity, rapid production, cost-effectiveness due to reduced material consumption, and the ability to produce complex structures [31,32,33].

Combining 3D printing technology with the intrinsic characteristics of PLA has shown high potential for creating intricate biomedical devices based on computer designs that use patient-specific anatomical data [34]. Three-dimensional printing using PLA is used to create special devices, improve implants, and make better scaffolds for tissue engineering, diagnostic tools, and delivering medicines [35,36]. In addition, the combination of PLA and FDM plays a vital role in the other areas of bioindustry. This is practical for creating lab equipment, teaching tools, surgical devices (fixation rods, plates, pins, screws, sutures, etc.), and agricultural instruments [37].

More widely, 3D printing of PLA, combined with other materials, results in composites vastly used in different industries, specifically aerospace and aviation fields. In these applications, PLA is used as a matrix. The advantage of this application is the final product’s increased strength and modulus [38].

### 4.2. Degradation Causes and Degrees of Effect

The effect of time on the mechanical properties of PLA, particularly its stress–strain curve, can depend on several factors, including the environmental conditions to which the material is exposed, such as exposure to UV light, moisture, other external factors, and the rate of loading [20,21]. That is, over time, PLA can undergo various degradation processes, such as hydrolysis, oxidation, and thermal degradation, all of which can cause the molecular weight of PLA to change.

In a study, dogbone samples 3D printed from PLA were left for around a month in an environment at 23 °C (room temperature). The resulting tensile stress–strain curves revealed that the natural degradation of PLA caused a tensile failure strain reduction by around 34% after 24 days [39]. This value was much greater than what was seen in our study, which showed a decline in failure strain of 5.3% after 45 days. This difference can be due to variations in material properties, environmental conditions, and experimental setups. PLA filament density was 12% higher in our study, and temperature and UV radiation, due to the weathering environment, were lower in our experiments, showcasing the significant contribution of these factors to the degradation rate of PLA. Notably, the difference in cross-head speed (50 mm/min in the mentioned study) might have also influenced the results, potentially leading to overestimation. To reduce the sources of variability in the results, it is recommended to prepare standardized procedures of degradation testing for different industrial use cases and climates.

The reason why each environmental condition affects the mechanical behavior of PLA is described in the following subsections.

#### 4.2.1. Moisture

One of the most common factors that can affect the mechanical properties of PLA over time is exposure to moisture. PLA is known to absorb moisture from the environment, which can reduce its mechanical properties. Numerous studies have argued that moisture can degrade PLA by breaking the ester bonds, reducing the material’s strength, stiffness, and molecular weight [30,40,41,42]. This is because moisture breaks down the polymer chains and reduces the intermolecular forces that hold the polymer together [43]. As a result, the material becomes less resistant to deformation and stress, so it is likely to become more brittle.

Moisture absorption can also affect the dimensional stability of PLA. As moisture is absorbed, the material can swell, which changes its size and shape. This can be a concern in applications where precise dimensional tolerances are critical. Therefore, it is essential to store and handle PLA properly to prevent moisture absorption and ensure the material’s optimal performance.

#### 4.2.2. Temperature

Thermal degradation is another degradation process. PLA can slowly degrade even under room temperature. In a study by Karamanlioglu, M., et al., when samples were kept at room conditions of 20 ± 2 °C with 40 ± 10% relative humidity in the dark for four years, their fracture strain and tensile strengths exhibited 82% and 34% loss, respectively [44]. PLA can particularly undergo thermal degradation at high temperatures, causing the material to break down and lose its mechanical properties. The thermal degradation of PLA can lead to a reduction in its molecular weight. PLA’s strength decreases as the temperature increases [45].

At low temperatures, PLA is rigid and brittle, and its stress–strain curve is similar to that of a ceramic material. As the temperature increases, PLA undergoes a transition towards softening and enhanced ductility, and its stress–strain curve begins to resemble that of a typical thermoplastic material.

At high temperatures, the stress–strain curve of PLA can also exhibit a different shape, with a more gradual transition from the elastic to plastic regions due to the increased ductility of the material. This can lead to a lower ultimate strength, but a higher elongation at break. Ductile polymers usually possess a well-defined yield point with a significant strain, often around 5–10 percent, due to their semi-crystalline state. In contrast, most amorphous and glassy polymers tend to be brittle and rupture at relatively low strain levels. However, these effects can be controlled by adding plasticizers or adjusting the temperature [45].

#### 4.2.3. UV Exposure

Another factor that can affect the mechanical properties of PLA over time is exposure to UV light. UV light can break down the chemical bonds in PLA. To demonstrate the effect of UV light exposure on the stress–strain curve of PLA, a tensile test on samples exposed to different amounts of UV light was conducted in [46]. The sample exposed to UV light exhibits lower strength and lower elongation at break compared to the unexposed sample. This suggests that UV light exposure has caused some degradation of the material, resulting in reduced mechanical performance [46].

It is worth noting that the degree of degradation observed in PLA due to UV light exposure can depend on various factors, such as the intensity and duration of exposure, the wavelength of the UV light, and the specific formulation of the PLA material.

In another work, after 24 h of exposure to UV-B irradiation, PLA tensile strength and compression strength declined by 5.3% and 6.3%, respectively [47]. Furthermore, the aging of PLA under sterilizing UV-C radiation was examined in [48]. Both tensile and compressive tests were conducted and showed a 9.1% loss in tensile strength compared to that in the control group. Necking strain also underwent a decline. In agreement with tensile strength, compressive strength decreased by 13.1% after exposure. Although the samples all revealed plastic behavior under compression, failure strain exhibited a decrease [48].

In another work, 3D printed products were exposed to UVA-340 lamps, which are good representatives of sunlight in the wavelength region between 365 nm and 295 nm, for 0, 5, 10, and 20 days [49]. Over a 20-day period, UV exposure caused the ultimate tensile stress of the samples to decline from 26.5 MPa to 14.7 MPa.

#### 4.2.4. Combined Effects of Moisture, Temperature, and UV Exposure

Understanding the interplay between environmental factors is crucial for predicting PLA degradation, as the combined effects of moisture, temperature, and UV exposure can significantly accelerate the process through complex interactions. Notably, the combination of humidity and elevated temperatures drastically accelerates the hydrolysis of PLA’s ester bonds [46]. For example, it has been shown that in a humid environment, an increase in temperature accelerates hydrolysis, leading to a sharp degradation rate at 57 °C compared to 23 °C, and it has been shown that a temperature increase beyond 69 °C has no additional degradation effect due to PLA saturation [50]. Similarly, in combination, UV radiation, generating free radicals in PLA, and temperature variations that increase the mobility of these radicals and the interaction with water molecules in humidity exacerbate PLA thermal and hydrolysis degradation more than either factor alone [51,52]. These combinations lead to rapid chain scission and a significant decline in mechanical properties [53].

### 4.3. Degradation Impact on Other Materials and 3D Printing Techniques

The question that arises here is whether or not changing the material and 3D printing technology can help mitigate the degradation of a manufactured part. In the following subsections, we present an overview of the effect of different environmental factors on the degradation of other materials or materials that are manufactured using other 3D printing technologies. The environmental factors that will be described in the following can be compared to the environmental factors mentioned for PLA in Section 4.2, and hence one can choose the appropriate material and manufacturing technology based on their application and environmental conditions.

#### 4.3.1. FDM Method with Other Polymers

There are other polymers widely used as FDM filaments that can be highly affected by environmental factors. The hygroscopic nature of nylons in 3D printing is also influenced by temperature and humidity, with higher humidity leading to increased moisture absorption and noticeably reduced mechanical properties, in particular, yield strength, Young’s modulus, and failure strain [54]. ABS (Acrylonitrile Butadiene Styrene) is a widely used thermoplastic known for its high durability, and unlike nylons, it shows lower moisture absorption even compared to PLA at room temperature [55]. ABS parts created through injection molding may exhibit different properties compared to ABS parts created through 3D printing [56].

Speaking of strength, in a study monitoring the impact of UV radiation, high temperature, high humidity, temperature fluctuations, and weather conditions on structures made by FDM technology, several materials were considered: PLA, PETG (Polyethylene Terephthalate Glycol), ABS, and ASA (Acrylonitrile Styrene Acrylate) [57]. Based on the observations in this study, in undegraded samples, PLA appears to have the highest ultimate strength. However, unlike other materials, high simultaneous humidity and temperature in a condensation chamber (100% humidity and temperature of 55 °C), temperature cycles (temperature variation between −18 °C and 21 °C), and being in outdoor environmental conditions (humidity from 30% to 97%, temperature from −5 °C to 10 °C, and solar radiation minimum intensity of $120{\;\mathrm{Wm}}^{-2}$) adversely affected PLA. This polymer is also limited in use at higher temperatures due to its softening temperature and reduced shape stability. On the other hand, under condensation chamber conditions and temperature cycles, PETG demonstrates higher tensile strength levels than PLA. In fact, the ultimate strength of PETG even undergoes a slight increase under all tested factors. PETG also exhibits the highest ductility among the materials tested. ABS, however, has the lowest tensile strength values, which witness a slight degradation under all the above-mentioned environmental factors. ASA material, despite having lower strengths compared to PLA and PETG, exhibits the least variation in properties under individual environmental factors, making it a stable option with higher hardness and resistance to higher temperatures than PETG when a product is exposed to the mentioned conditions [57].

In general, PLA exhibits promising behavior due to its bio-inspired nature, and it is widely used due to its recyclability or reusability. The biodegradability of PLA reduces long-term waste, typical for 3D printed objects [58]. Similarly, novel materials such as Polyhydroxyalkanoates (PHAs) and Polycaprolactone (PCL) demonstrate an acceptable biodegradable nature [59]. In contrast, ABS, nylon, and ASA, as petroleum-based plastics, cause large carbon footprints. ABS, while durable, emits significant amounts of toxic fumes, such as styrene, during printing and requires proper ventilation. Its production process is also more energy-intensive, contributing to its larger environmental inefficiency. Therefore, ABS should be altered wherever applicable [60]. Although ABS, PETG, and nylon can all be recycled under certain conditions [61], PETG shows a higher potential to be recycled and produces fewer toxic fumes [62]. However, it is still petroleum-based and non-biodegradable. PETG provides better environmental stability but lacks the biodegradability of PLA, which is a sustainable choice for controlled conditions.

#### 4.3.2. Other 3D Printing Methods

Degradation is not confined to the products made by FDM technology. As for the degradation of SLA (Stereolithography) products, UV aging was conducted on three types of resins (tough, flexible, and strong) in [63], all of which exhibited a decline in their elongation at break, tensile, and impact strength, with a slight increase in Young’s modulus. Therefore, prolonged exposure to UV for resins used in SLA makes them brittle [64].

The degradation of SLS printed products due to aging or reusing has been observed to lead to a significant reduction in tensile strength, Young’s modulus, and elongation at break. In a study [65] conducted on parts made from different semi-crystalline polymer polyamide 12 (PA12) powders, the degradation process was assessed. For samples that underwent reuse, there was a notable decline in tensile strength. Specifically, the first sample type (where the powder was reused 10 times) exhibited a reduction from an initial value of 35 MPa to 2 MPa, while the second sample type (where the powder was reused 8 times) exhibited a decrease from 31.65 MPa to 20.45 MPa. Similar trends were observed for Young’s modulus, where the first sample type’s value dropped from 2000 MPa to 300 MPa, and the second sample type’s value decreased from 1275.67 MPa to 1034.5 MPa. Elongation at break also exhibited significant decreases for the reused samples. The first sample type saw a decrease in elongation at break from an initial value of 5.3% to 0.2%, while the second sample type exhibited a reduction from 6.24% to 4.55%. This comparative analysis was conducted between new powders and powders that had been reused 10 and 8 times.

### 4.4. Prevention and Mitigation of PLA Degradation

#### 4.4.1. Methods

As the significance of failure strain on the dynamic response of sandwich structures made up of PLA was observed in this study, to enhance the behavior of structures under impact loading, it is suggested to focus on improving the failure strain limit more than the tensile strength. It has been shown the application of different thermal treatments to PLA can considerably increase its ductility while having minimal impact on the ultimate stress [41]. Annealing and temperature treatment together improve the mechanical properties of PLA [66]. Adding small amounts of fillers and other materials to PLA can increase tensile strength and failure strain in undegraded PLA, while also enhancing material degradation/loss behavior after degradation. Nevertheless, they have conflicting results for higher percentages of fillers. Therefore, the addition of fillers requires thorough investigation before implementation [55].

In general, various strategies can lower PLA’s degradation rate in real-world applications, ranging from product design considerations to environmental protection measures. Design adjustments, such as using thicker cross-sections in stress-prone areas [67] and ensuring even stress distribution, can enhance product durability with minimal additional costs. More specialized strategies include material modifications through thermal treatments, additives, and, as mentioned before, fillers. Surface coatings and controlled storage conditions further protect the product against environmental degradation.

Incorporating moisture barriers and co-polymerizing lactide with hydrophobic monomers can significantly enhance the hydrolysis resistance of PLA [68,69]. Techniques such as dip-coating with hydrophobic silica particles or using initiated chemical vapor deposition (iCVD) to coat 3D printed PLA objects with hydrophobic polymers are highly effective. Blending PLA with heat-resistant polymers such as polyhydroxybutyrate (PHB) or adding heat stabilizers can mitigate temperature-related degradation. For instance, PLA/poly(3-hydroxybutyrate-co-3-hydroxyvalerate) composites exhibit 16% higher strength and 15 times higher ductility compared to pure PLA [69].

Including UV absorbers in PLA formulations, such as diethyl ether extractives from Phoebe zhennan wood [70], or applying UV-protective coatings, particularly sustainable multifunctional bioderived ones, can protect the material from photodegradation [71]. A study showed that adding high content of 15 wt % cellulose nanocrystal-zinc oxide (CNC-ZnO) hybrids to PLA can block around 85% and 96% of UV-A and UV-B effects, respectively [72]. Furthermore, a coating prepared from chitosan (CS), tannic acid (TA), and phytic acid (PA) (PA@TA-CS) could block around 99% of UV light [73].

Controlled storage conditions, with regular monitoring of variations in humidity, temperature, and UV exposure, can also be very beneficial. A dry and UV-blocking environment with a stable, low temperature can significantly decrease the degradation rate [74]. It has been shown that lactide degrades faster in a natural environment than in an argon-filled glove box, emphasizing the importance of storage in a low-oxygen and low-moisture environment for maintaining the quality of PLA [75]. This can be achieved using vacuum-sealed bags with desiccators [76]. Additionally, according to another study, freezing PLA filaments at −24 °C or lower can prevent aging for up to nine months, or potentially indefinitely, without causing mechanical damage, using polyethylene terephthalate (PET) zip-bags, making it a cost-effective, simple, and highly effective anti-aging procedure [29].

#### 4.4.2. Costs of Improvement Techniques

Each method’s cost-effectiveness should be evaluated based on specific application requirements, and a trade-off between the cost and the importance of protection must be considered. Simpler methods such as thermal treatments (e.g., annealing) [77] and the addition of low-percentage fillers, such as natural fibers, to PLA are generally economical and recommended when PLA products are required to showcase durability under environmental conditions [78]. This filler incorporation approach requires homogeneous distribution [79]. Additionally, optimizing and maintaining environmental conditions include initial setup costs for the infrastructure. Despite this dependency on the application, the basic initial steps to keep a dry, cool, and UV-protected environment, if applicable, are considered low-cost with respect to their significance.

Surface coatings such as UV absorbers and moisture barriers vary in cost. Processes such as all-dry iCVD efficiently reduce costs associated with the traditional solution-based iCVD method [80]. On the other hand, maintaining transparency and mechanical strength in UV-enhanced PLA can be costly [81].

Bio-based (e.g., lignin-based) materials used in coatings or additives generally offer cost-effective alternatives across various material and surface modification methods [82]. However, blending PLA with other biopolymers can involve higher costs compared to synthetic polymers, including biopolymers with reinforcing agents such as nanofillers and active agents [83]. The choice of prevention method should therefore be tailored depending on application conditions, and it can be based on one of the noted various degradation mitigation approaches or a combination of them.

### 4.5. Applications

By thoroughly evaluating and monitoring the mechanical behavior of lattice and porous designs within sandwich structures, particularly their impact response, engineers gain insights into degradation mechanisms and predict their durability. This knowledge is crucial for developing sandwich panels that maintain their energy absorption and strength characteristics while ensuring their effectiveness in impact-exposed applications. Ultimately, this research contributes to designing 3D printed structures that can withstand varying environmental conditions and maintain their intended functionality and safety.

## 5. Conclusions

This study explored the detrimental effects of environmental factors on the mechanical properties (yield strength and failure strain) of PLA and their influence on the low-velocity impact response of sandwich structures with different lattice cores (auxetic, NTP gradient, PTN gradient, and honeycomb). The research compared key aspects of sandwich panels with four core types (auxetic, gradient negative-to-positive (NTP), gradient positive-to-negative (PTN), and honeycomb) including impactor penetration depth, crushing area extent, stress distribution, energy absorption, and force–displacement curves. Based on the comparisons, the following key findings were observed for degraded sandwich panels:Auxetic structures exhibited higher resilience compared to honeycomb structures in both degraded and undegraded conditions. Notably, the weakest auxetic core had 24% higher penetration compared to its strongest case, while the weakest honeycomb core showed a 36% increase compared to its strongest version.Interestingly, reducing yield strength by 60% with constant failure strain had minimal effect on energy absorption for all structures except auxetic ones.A 60% reduction in failure strain (more brittle material) caused a significant (around 50%) decrease in energy absorption across most core types, with honeycomb cores experiencing the most significant drop (60% decrease). Auxetic structures showed a smaller reduction in energy absorption (~35%) compared to other core types.Under moderate degradation (yield strength and failure strain at 60% and 80% of reference values), the gradient cores maintained similar peak forces (around 800 N) to the auxetic core (840 N) while exhibiting superior energy absorption (3.5 J vs. 2.5 J in auxetic). This suggests that gradient structures can balance strength and energy absorption under degradation.Failure strain had a more significant influence on overall displacement compared to yield strength for all core types.Despite potential nuances in deformation and penetration behavior that require further investigation, gradient structures show promise for maintaining strength, superior energy absorption, and potentially different modes of energy dissipation across several degradation levels.

## Figures and Tables

**Figure 1 materials-17-03674-f001:**
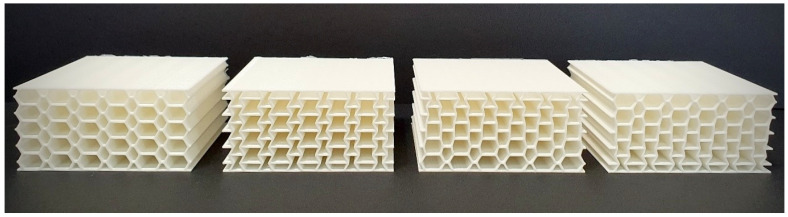
The printed samples. From left to right: honeycomb, auxetic, NTP gradient, and PTN gradient sandwich panels.

**Figure 2 materials-17-03674-f002:**
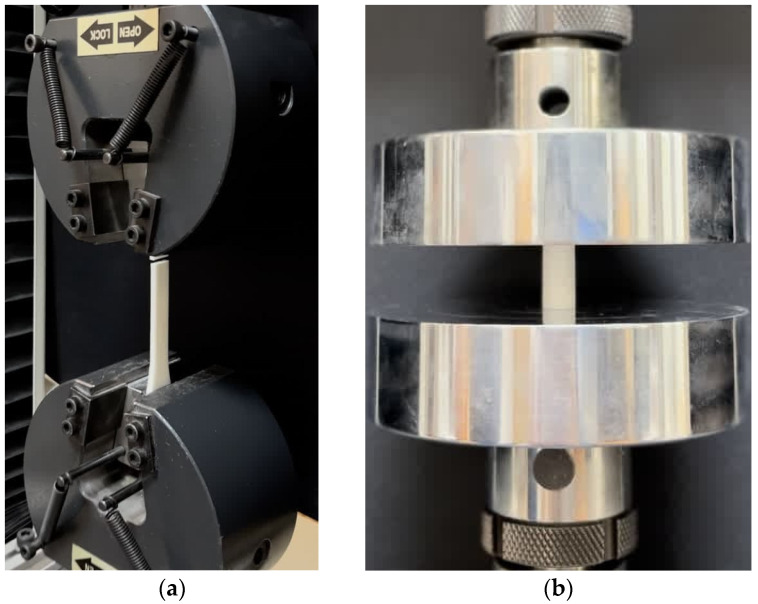
Quasi-static tests: (**a**) tensile, (**b**) compressive setups.

**Figure 3 materials-17-03674-f003:**
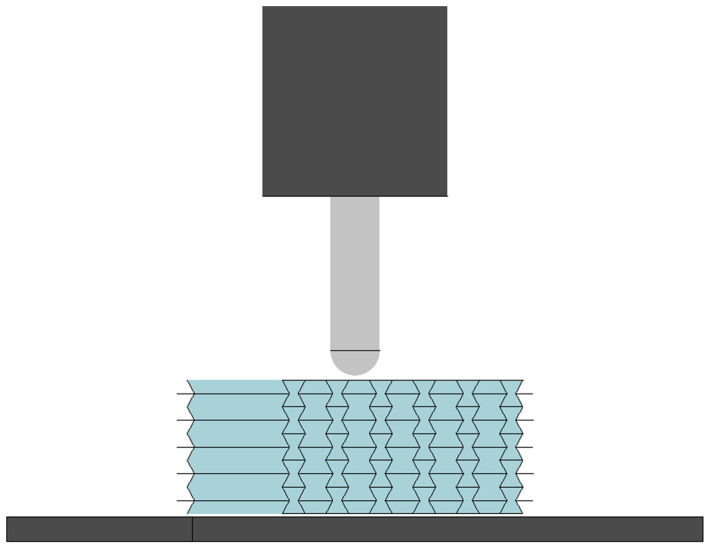
Finite element model for the low-velocity impact test.

**Figure 4 materials-17-03674-f004:**
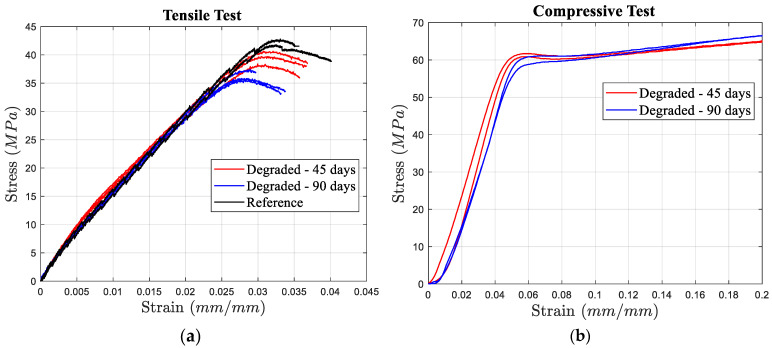
(**a**) Tensile and (**b**) compressive stress–strain curves of the specimens tested 2 days, 45 days, and 90 days after manufacturing.

**Figure 5 materials-17-03674-f005:**
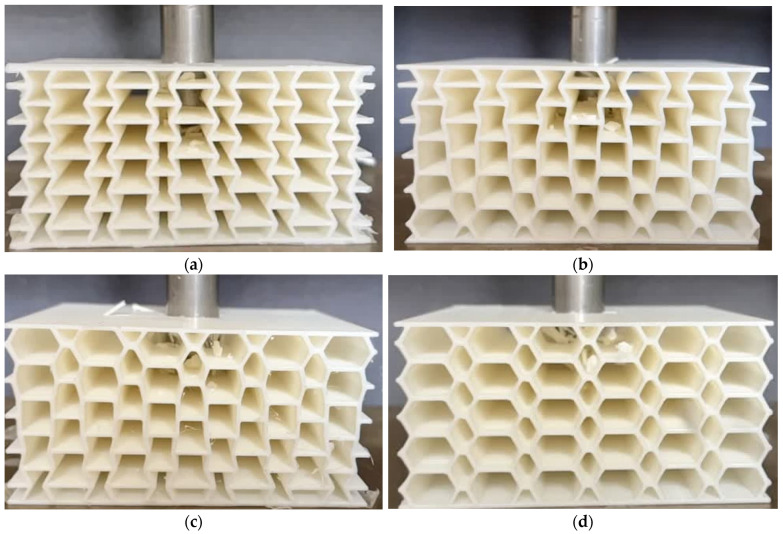
Drop-weight impact of sandwich panels with (**a**) auxetic, (**b**) NTP gradient, (**c**) PTN gradient, and (**d**) honeycomb cores.

**Figure 6 materials-17-03674-f006:**
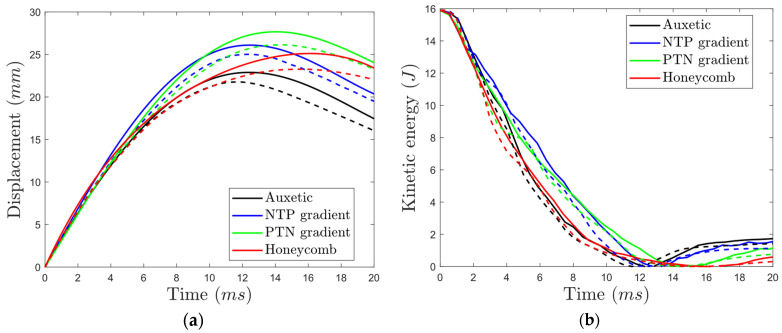
Drop-weight impact results for undegraded cores: (**a**) displacement–time and (**b**) kinetic energy–time curves of the impactor (solid lines: experimental data, dashed lines: FE model).

**Figure 7 materials-17-03674-f007:**
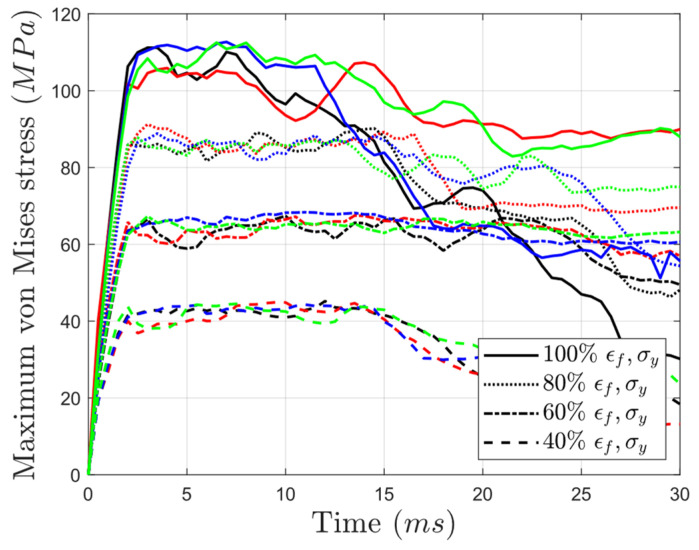
Variation in the maximum von Mises stress with time for structures made up of PLA having yield strengths and failure strains equal to 40%, 60%, 80%, and 100% of those in the as-printed PLA (black: auxetic, red: honeycomb, blue: NTP gradient, green: PTN gradient).

**Figure 8 materials-17-03674-f008:**
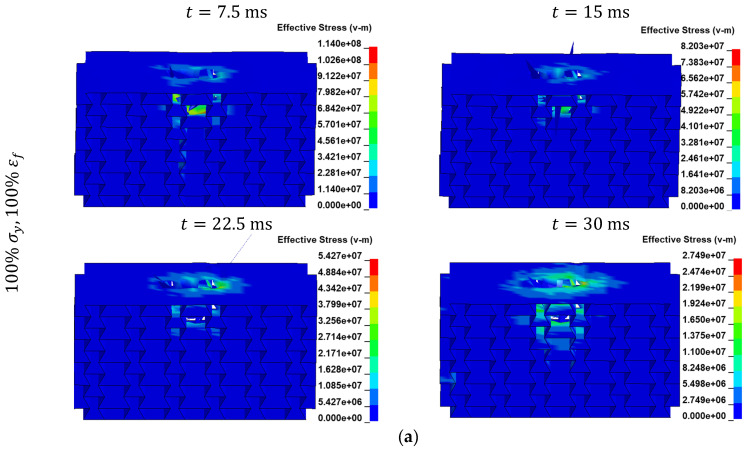
von Mises stress distribution for the auxetic structure made up of PLAs having yield strengths and failure strains equal to (**a**) 100%, (**b**) 80%, (**c**) 60%, and (**d**) 40% of those in the as-printed PLA at t=7.5 ms, t=15 ms, t=22.5 ms, and t=30 ms.

**Figure 9 materials-17-03674-f009:**
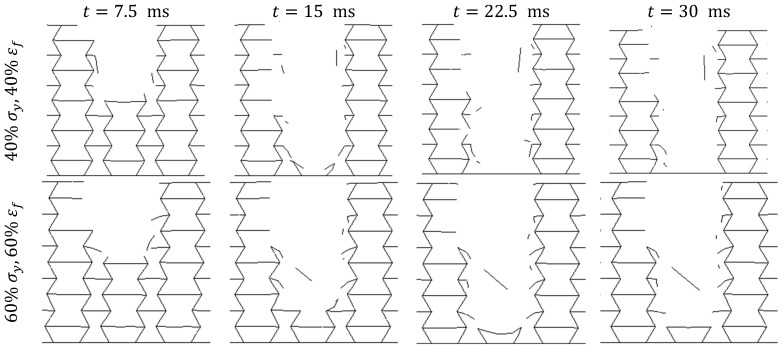
Cross-sectional views of the deformation of the impacted region of the auxetic structures made up of PLAs having yield strengths and failure strains equal to 40%, 60%, 80%, and 100% of those in the as-printed PLA at t=7.5 ms, t=15 ms, t=22.5 ms, and t=30 ms.

**Figure 10 materials-17-03674-f010:**
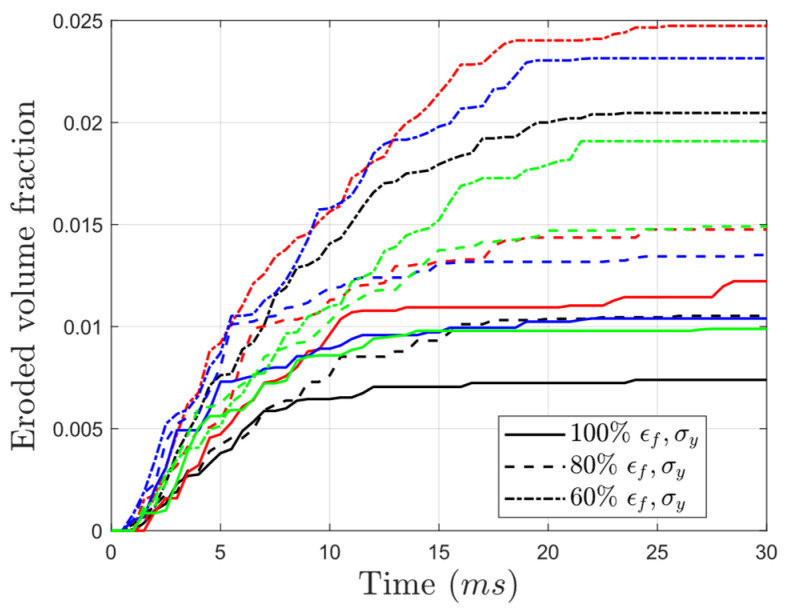
Variation in eroded volume fraction with time for structures made up of PLA having yield strengths and failure strains equal to 60%, 80%, and 100% of those in the as-printed PLA (black: auxetic, red: honeycomb, blue: NTP gradient, green: PTN gradient).

**Figure 11 materials-17-03674-f011:**
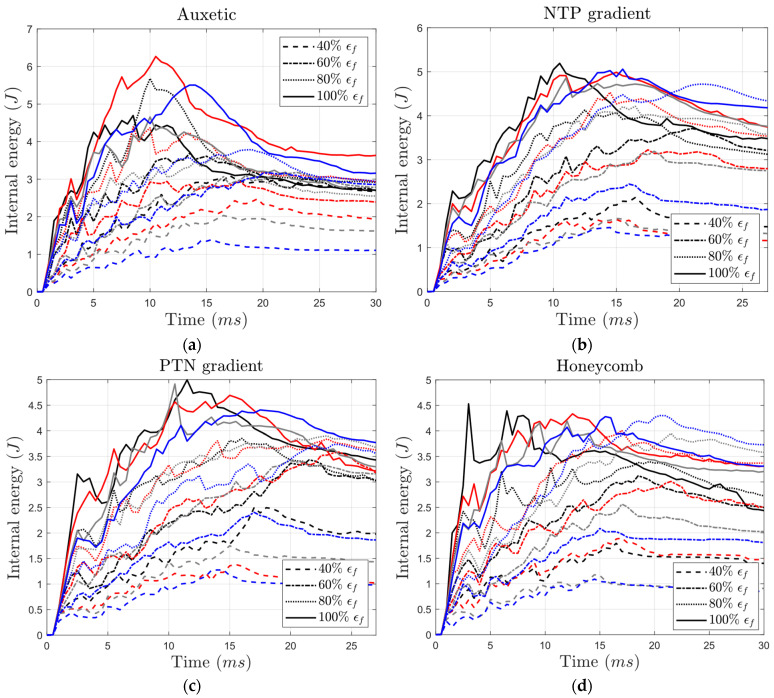
Variation in the internal energy over time for (**a**) auxetic, (**b**) NTP gradient, (**c**) PTN gradient, and (**d**) honeycomb core types having different levels of yield stress and failure strain for core material (black: 100% $\sigma_{y}$, red: 80% $\sigma_{y}$, gray: 60% $\sigma_{y}$, blue: 40% $\sigma_{y}$).

**Figure 12 materials-17-03674-f012:**
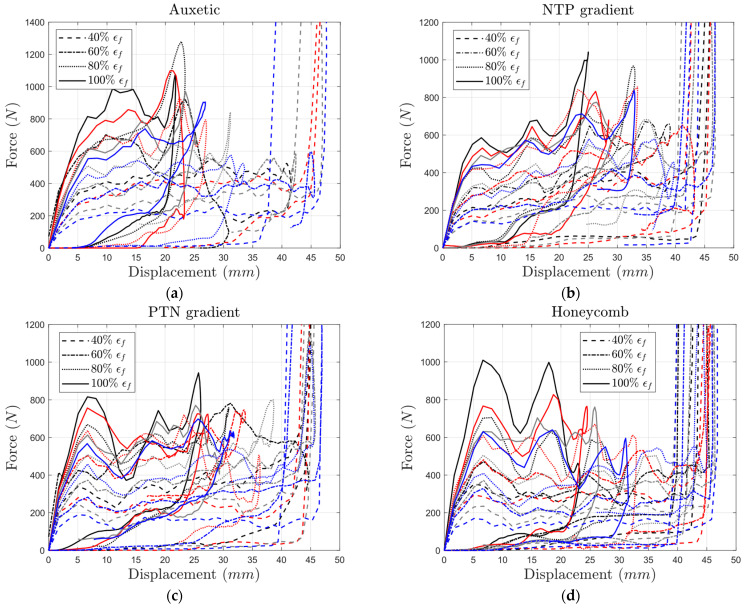
Variation in contact force with impactor’s displacement for (**a**) auxetic, (**b**) NTP gradient, (**c**) PTN gradient, and (**d**) honeycomb core types with different yield strength and failure strain levels (black: 100% $\sigma_{y}$, red: 80% $\sigma_{y}$, gray: 60% $\sigma_{y}$, blue: 40% $\sigma_{y}$).

**Figure 13 materials-17-03674-f013:**
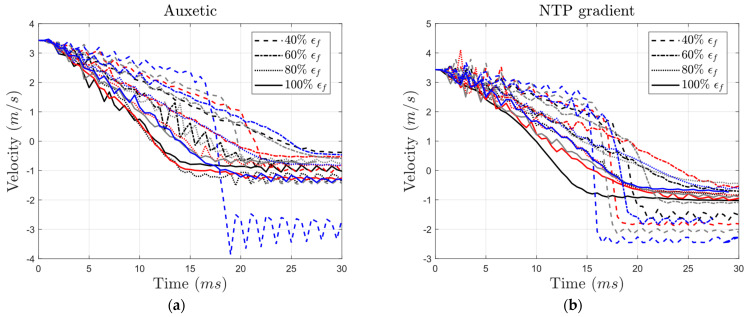
Variation in impactor’s velocity with time for (**a**) auxetic, (**b**) NTP gradient, (**c**) PTN gradient, and (**d**) honeycomb core types with different yield strength and failure strain levels (black: 100% $\sigma_{y}$, red: 80% $\sigma_{y}$, gray: 60% $\sigma_{y}$, blue: 40% $\sigma_{y}$).

**Figure 14 materials-17-03674-f014:**
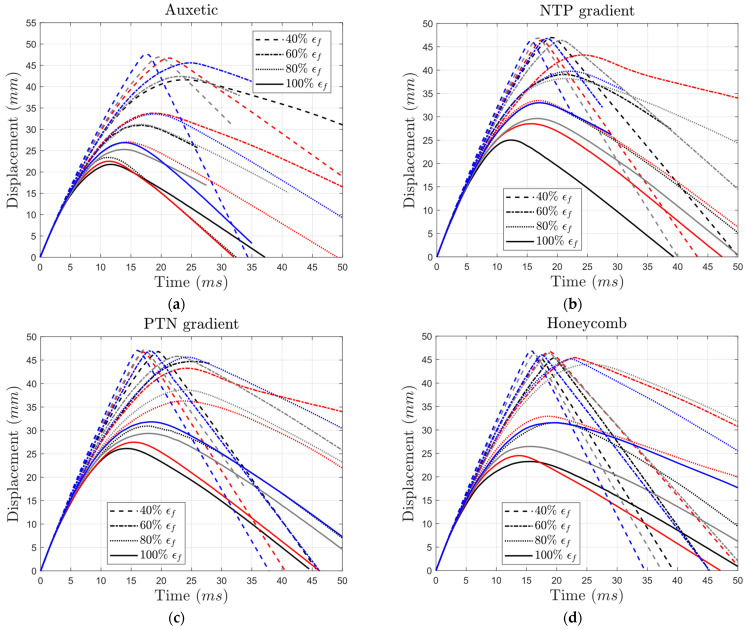
Variation in impactor’s displacement over time for (**a**) auxetic, (**b**) NTP gradient, (**c**) PTN gradient, and (**d**) honeycomb core types with different yield strength and failure strain levels (black: 100% $\sigma_{y}$, red: 80% $\sigma_{y}$, gray: 60% $\sigma_{y}$, blue: 40% $\sigma_{y}$).

**Figure 15 materials-17-03674-f015:**
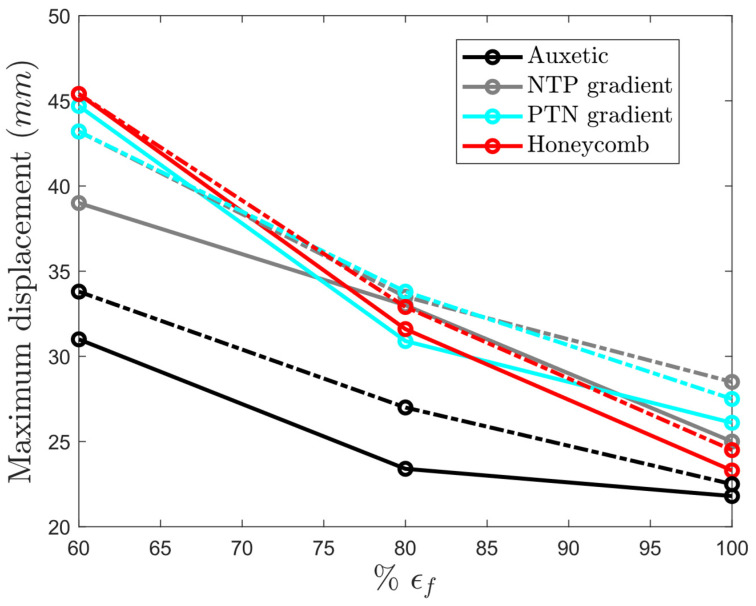
Maximum impactor’s displacement per failure strain for yield strength of 100% (solid lines) and 80% (dashed-dotted lines) in all structures.

**Table 1 materials-17-03674-t001:** Mechanical properties of the materials used.

	PLA (Core)	Steel (Impactor)	Extra Weight
Mass (kg)	0.126	0.0863	2.99
Density (kg/m^3^)	1200	7800	10,610
Young’s modulus (GPa)	1.3	200	200
Poisson’s ratio	0.35	0.3	0.3
Yield strength (MPa)	37	-	-
Tangent modulus (MPa)	0.79	-	-
Failure strain	0.04	-	-

## Data Availability

The original contributions presented in the study are included in the article/Appendix A, further inquiries can be directed to the corresponding author.

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
