# Peer review of "Effect of Degradation of Polylactic Acid (PLA) on Dynamic Mechanical Response of 3D Printed Lattice Structures"

_materials, 2024, doi:10.3390/ma17153674_

Round 1

Reviewer 1 Report

Comments and Suggestions for Authors

The study “Effect of Degradation of Polylactic Acid (PLA) on Static and Dynamic Mechanical Response of 3D Printed Lattice Structures”

1.     I think the first paragraph in the abstract is not necessary because it is reproduced in the introduction.

2. Including a review of auxetic materials in the introduction.

3.     In compression tests, why is the complete piece not tested? as in the Drop-weight impact of sandwich panels

4.     Fig. 8 and 9. I think that images should be selected since it is not practical to appreciate the behavior. Maybe include a table about your observations.

5.     Fig 11 and 12, it is difficult to understand the behavior with too much information in the figures.

Author Response

Please see the attached response file

Reviewer 2 Report

Comments and Suggestions for Authors

Dear Authors, I read with great interest your article discussing the prevalence, degradation causes, and impacts of degradation on 3D-printed PLA (polylactic acid) products, with a particular focus on additive manufacturing methods and the mechanical properties of PLA over time. The degradation of other polymeric materials and the resilience of different 3D-printed structures under various environmental conditions are also explored. I must acknowledge that the article has undeniable strengths, covering a wide range of topics related to PLA degradation, including environmental causes such as humidity, temperature, and UV exposure. The use of references to existing studies provides a solid foundation and enhances the credibility of the presented information. The sections addressing the causes of degradation (humidity, temperature, UV) offer a good overview but lack depth in discussing how these factors interact with each other and with the material itself.

It is necessary to provide a more in-depth discussion on how humidity, temperature, and UV exposure can simultaneously interact to influence PLA degradation. For instance, the combined effect of high humidity and temperature could be different from the effect of each factor individually. Although the article discusses methods to improve the mechanical properties of PLA (such as thermal treatments and the addition of fillers), it does not provide concrete proposals on how to mitigate degradation during practical use. It is suggested to include a section dedicated to practical strategies for mitigating PLA degradation in real-world applications. This could include suggestions for product design, the use of protective coatings, or optimal storage conditions. Insufficient emphasis is placed on the reproducibility of experiments and the variability of results obtained from different laboratories. Therefore, it is recommended to include a discussion on the reproducibility of the cited studies and the possible sources of variability in the results, and to provide recommendations on how to standardize experimental procedures to achieve more consistent results. The article does not address the costs associated with implementing techniques to improve PLA properties and does not discuss the environmental sustainability of using PLA compared to other materials. It would be appropriate to include a section discussing production costs, the costs of improvement techniques, and the environmental sustainability of using PLA compared to other commonly used 3D printing materials. In conclusion, the article provides a useful and well-documented basis on the degradation of 3D-printed PLA but can be improved in several key areas. A more detailed comparative analysis, a deeper discussion of the interactions between degradation factors, and concrete proposals for mitigating degradation would greatly enhance the quality and utility of the article, which is already a commendable scientific contribution. Additionally, greater precision in experimental details and a discussion on costs and sustainability would add significant value to the work. I encourage you to consider these suggestions to improve the scientific rigor and practical relevance of your study, for which I commend you for the excellent elaboration.

Author Response

Please kindly see the attached response file.

Round 2

Reviewer 1 Report

Comments and Suggestions for Authors

No more comments